# Small-Molecule Inhibitors of TIPE3 Protein Identified through Deep Learning Suppress Cancer Cell Growth In Vitro

**DOI:** 10.3390/cells13090771

**Published:** 2024-04-30

**Authors:** Xiaodie Chen, Zhen Lu, Jin Xiao, Wei Xia, Yi Pan, Houjun Xia, Youhai H. Chen, Haiping Zhang

**Affiliations:** 1Center for Cancer Immunology, Institute of Biomedicine and Biotechnology, Shenzhen Institute of Advanced Technology, Chinese Academy of Sciences, Shenzhen 518055, China; xd.chen@siat.ac.cn (X.C.); zhen.lu@siat.ac.cn (Z.L.); hj.xia@siat.ac.cn (H.X.); 2University of Chinese Academy of Sciences, Beijing 100049, China; 3Faculty of Synthetic Biology and Institute of Synthetic Biology, Shenzhen Institute of Advanced Technology, Chinese Academy of Sciences, Shenzhen 518055, China; 52264300064@stu.ecnu.edu.cn (J.X.); weixia@vip.126.com (W.X.); yi.pan@siat.ac.cn (Y.P.); 4Faculty of Pharmaceutical Sciences, Shenzhen University of Advanced Technology, Shenzhen 518055, China

**Keywords:** TIPE3, small-molecule compounds, anti-tumor, deep learning

## Abstract

Tumor necrosis factor-α-induced protein 8-like 3 (TNFAIP8L3 or TIPE3) functions as a transfer protein for lipid second messengers. TIPE3 is highly upregulated in several human cancers and has been established to significantly promote tumor cell proliferation, migration, and invasion and inhibit the apoptosis of cancer cells. Thus, inhibiting the function of TIPE3 is expected to be an effective strategy against cancer. The advancement of artificial intelligence (AI)-driven drug development has recently invigorated research in anti-cancer drug development. In this work, we incorporated DFCNN, Autodock Vina docking, DeepBindBC, MD, and metadynamics to efficiently identify inhibitors of TIPE3 from a ZINC compound dataset. Six potential candidates were selected for further experimental study to validate their anti-tumor activity. Among these, three small-molecule compounds (K784-8160, E745-0011, and 7238-1516) showed significant anti-tumor activity in vitro, leading to reduced tumor cell viability, proliferation, and migration and enhanced apoptotic tumor cell death. Notably, E745-0011 and 7238-1516 exhibited selective cytotoxicity toward tumor cells with high TIPE3 expression while having little or no effect on normal human cells or tumor cells with low TIPE3 expression. A molecular docking analysis further supported their interactions with TIPE3, highlighting hydrophobic interactions and their shared interaction residues and offering insights for designing more effective inhibitors. Taken together, this work demonstrates the feasibility of incorporating deep learning and MD simulations in virtual drug screening and provides inhibitors with significant potential for anti-cancer drug development against TIPE3−.

## 1. Introduction

The TIPE (tumor necrosis factor-α-induced protein 8-like, or TNFAIP8L) family of proteins comprises recently identified regulators of immunity and tumorigenesis. They are recognized as exclusive transfer proteins responsible for the lipid second messengers PtdIns(4,5)P2 (PIP2) and PtdIns(3,4,5)P3 (PIP3) [1,2,3]. As a member of the TIPE family, TIPE3 is predominantly expressed within the cytoplasmic compartments of epithelial-derived cells and exhibits notably high expression levels across a wide array of human carcinoma cell lines [4]. TIPE3 was clarified to transfer, present, and protect lipids such as phosphoinositides [5,6], which are involved in the regulation of cell signaling [7,8].

In 2014, Fayngerts et al. reported for the first time the carcinogenic function of TIPE3, evidenced by a significant deceleration in tumorigenesis upon TIPE3 knockout, while overexpression accentuated it [5]. The aberrant activation of the PI3K-AKT signaling network is one of the most frequent events in human cancer and serves to disconnect the control of cell growth, survival, and metabolism from exogenous growth stimuli [9,10]. TIPE3 can modulate the spatiotemporal distribution of lipid second messengers by stripping PtdIns(4,5)P2 and PtdIns(3,4,5)P3 from the lipid bilayer and shuttling them between the cell membrane and the cytoplasm; in turn, this regulates lipid metabolism and the activation of the PI3K/AKT and MEK-ERK signaling pathways [5]. Subsequently, researchers validated the oncogenic function of TIPE3 in a variety of human cancers [11,12,13,14,15,16]. In lung cancer, TIPE3 was found to be upregulated and could activate the AKT/ERK1/2-GSK3β-b-catenin/Snail pathway, enhancing the migration of cancer cells [12]. In gastric cancer, the overexpression of TIPE3 promoted GC cell growth and metastasis by upregulating the phosphorylation of PI3K and AKT [17]. Furthermore, inhibitors of the AKT and NF-kB pathways could block the proliferation and migration of breast cancer cells induced by TIPE3 overexpression [11]. Given that TIPE3 has an important and common regulatory role in the initiation and progression of multiple cancers, an effective broad-spectrum anti-tumor therapeutic strategy targeting TIPE3 using small-molecule inhibitors is revealed.

TIPE3 was first identified in 2008 as a TH domain-containing protein similar to TIPE2, a negative regulator of immunity and inflammation [2]. The crystal structure of TIPE3 revealed a hydrophobic pocket within the TH domain capable of binding phospholipids [5], most prominently PtdIns(4,5)P2, PtdIns(3,5)P2, PtdIns(3,4)P2, and PtdIns(3,4,5)P3. Importantly, mutation within the hydrophobic cavity disrupted this binding as well as the ability of TIPE3 to increase PtdIns(3,4,5)P3 levels and AKT activation. The disruption of this binding site also inhibited TIPE3 function. In addition, TIPE3 contains a unique N-terminal domain that is required for its activating functions; the deletion of this domain preserved its ability to bind phosphoinositides but resulted in a dominant negative, inhibitory phenotype akin to the function of TIPE2 [5].

Deep learning-based methods show promise for identifying TIPE3 inhibitors in a fast, accurate, and low-cost manner. The rapid advancement of deep learning has significantly influenced drug discovery [18,19,20], with deep learning-based methodologies becoming prevalent in the large-scale screening of potential drug candidates [21,22,23]. While numerous protein-ligand interaction prediction models have been proposed, such as Pafnucy [24], SE-OnionNet [25], onionNet [26], Kdeep [27], and OnionNet-2 [28], they are often hindered by their heavy reliance on docking conformations, resulting in slow performance. Additionally, the majority of current methods focus solely on affinity prediction, neglecting non-binding data entirely. However, there are also some non-complex dependent models, such as DeepDTA [29], GraphDTA [30], and DeepLPI [31]. Some models, such as PIGNet2, also consider negative data during training [32]. There are also many deep learning models for protein–ligand docking, such as AQDnet [33]. Inspired by these works, we developed several deep learning-based models for -protein–ligand interactions and built several pipelines for drug screening [34,35].

Moreover, molecular dynamics simulations play a crucial role in the quest for inhibitors with strong binding affinities for their respective targets [36]. Therefore, they also show promise for identifying inhibitors of TIPE3 in a slow but more accurate manner. Techniques like metadynamics [37] and funnel metadynamics [38] enhance the assessment of binding free energy landscapes, contributing to more accurate predictions. Established docking tools such as Autodock Vina [39] and Schrödinger [40] are instrumental in identifying the potential binding conformations of ligands, further aiding in the drug discovery process.

In this work, we constructed a TIPE3− ligand complex model leveraging the crystal structure of TIPE3, which enabled us to perform a large-scale virtual screening of the ZINC database to identify potential inhibitors. Of 58 shortlisted candidates, 6 were chosen for experimental validation. Notably, three of these compounds exhibited significant anti-tumor activities, including decreasing cell viability and proliferation, inhibiting cell migration, and enhancing apoptosis in tumor cells. Distinctively, two compounds from this subset selectively induced cell death in tumor cell lines while sparing human primary T cells, underscoring their therapeutic selectivity. The findings from our investigation underscore the utility and promise of adopting deep learning and molecular dynamics (MD) simulations in the drug discovery process, particularly for large-scale lead identification. The balance between accuracy and throughput achieved herein signifies a substantial advancement over traditional methods. Despite the scarcity of studies integrating large-scale deep learning with MD simulations in drug screening, our previous efforts in drug repurposing targeting RdRp and TIPE3 marked an initial foray into this domain [35,41]. However, the limited scope of the compound databases in those studies did not fully exploit the capabilities of this approach. By contrast, the current initiative entailed screening an excess of 10 million compounds, thereby establishing a new paradigm in the scope of drug discovery. This considerable expansion in scale from several thousands to millions of compounds propels this methodology to the forefront of innovative drug-screening strategies, laying the groundwork for subsequent explorations in this promising field.

## 2. Materials and Methods

### 2.1. Structural Modeling of TIPE3 and Compound Dataset

The 3D configuration of TIPE3 was acquired from the PDB database using the identifier 4Q9V [5]. TIPE3 and its ligand complex model were developed through the COFACTOR algorithm [42] applied within I-TASSER, leveraging a structure analysis and protein–protein interaction networks. By identifying amino acids located less than 1 nm from the ligand, we pinpointed the binding site. In our search for compounds with drug-like properties, we reviewed the ZINC15 database (2019 edition), ultimately compiling a virtual screening library consisting of 10,402,895 compounds.

### 2.2. Screening Pipeline

A hybrid approach combining deep learning and molecular simulation techniques was employed for conducting a virtual screening of the ZINC dataset, targeting TIPE3 [43]. This meticulous virtual screening process identified 58 compounds showing high potential for binding with TIPE3. Of these, six compounds were selected for a further validation of their effectiveness.

#### 2.2.1. Molecular Vector-Based Screening

A deep learning-based method, DFCNN (dense fully connected neural network), was used to conduct an initial drug screening [44,45]. DFCNN has used concise mol2vec [46] representation for the protein pocket and ligand, compared to many other methods. It is not dependent on the protein and ligand binding conformation, which greatly accelerates its prediction efficiency. Here, we used the training set’s mean and deviation values for the normalization of the testing data. The parameter settings for the DFCNN are the same as in our previous works, which examined repurposing a drug to inhibit the RdRp of SARS-CoV-2 [35] and screening a TIPE2 inhibitor [34].

#### 2.2.2. Structure-Based Drug Screening

DeepBindBC, a previously developed deep learning model, was utilized for structure-oriented drug discovery efforts [43]. In contrast to DFCNN, DeepBindBC processes not only physicochemical but also spatial data concerning the interfaces between proteins and ligands. This inclusion allows DeepBindBC to achieve higher accuracy, although it requires the availability of the protein–drug complex structure, which is procured using Autodock Vina (Autodock Vina software version 1.1.2, La Jolla, CA, USA).

Autodock Vina was engaged for the binding of targets with candidate ligands [39]. The binding site was identified based on the ligand’s position within a reference protein. The dimensions of the docking site were determined to be 2.5 nm on the x, y, and z axes, measured from the center of the pocket’s mass. To prepare for docking, AutoDock Tools transformed the PDB files into a PDBQT format [47]. The parameters of the docking process itself were meticulously set, with an exhaustiveness level of 8, a cap of 20 different binding modes to explore, and a maximum energy variance of 3 between them. The efficacy of Autodock Vina’s scoring and optimization mechanisms was thoroughly examined in an earlier publication.

DeepBindBC operates on a ResNet architecture fine-tuned using data from the PDBbind database [48]. In this system, interactions at the -protein–ligand interface are transformed into visual-like data points [43]. Through leveraging conformation data from cross-docking scenarios (in which proteins and ligands from separate experimental setups are docked) as negative examples in its training process, DeepBindBC effectively identifies compounds which are unlikely to bind.

Additionally, the Schrödinger Glide docking tool was employed for a subsequent round of compound evaluation, focusing on the compounds’ binding potential. Initially, ligands were optimized geometrically via the Ligprep tool. This optimization minimized each ligand’s energy through the OPLS 2005 force field and generated all possible ionization states at a pH of 7.4. Following this, a single low-energy three-dimensional configuration for was produced for each ligand, maintaining its original stereochemistry. Proteins were then prepared by adding hydrogen atoms and optimizing the entire system at a pH of 7.4, using the OPLS-u-2005 force field. The docking area was set up around the geometric midpoint of the protein-bound ligands, with scaling on all three axes at 2.6 Å. Glide docking was performed under default parameters, which included standard precision and flexible sampling without any restrictions.

#### 2.2.3. Force-Field-Based Screening

To further refine the drug discovery process, MD simulations based on force fields were conducted. In our research, we chose 58 complex formations involving compounds identified as potential candidates through prior deep learning analyses to undergo MD simulations individually. The capability to calculate the binding free energy was enabled by employing metadynamics simulations, which aid in determining the likelihood of binding interactions between a protein and ligand in a solution environment. Metadynamics facilitates this by implementing a biasing potential that aids in traversing the free energy landscape, focusing on particular collective variables of interest [49,50]. An exhaustive depiction of the MD simulations and subsequent metadynamics simulations is provided in Appendix A, detailing the process.

### 2.3. Cell Culture

HT-29, LoVo, TE-1, HCT 116, PANC-1, and Jurkat cells, verified by STR profiling, were sourced from the Cell Bank of the Chinese Academy of Sciences (Shanghai, China). A HeLa cell line (Cat. No. CL-0101) was procured from Procell Life Science & Technology, China. Tumor cell lines were cultured in DMEM (Gibco, Clinton, OK, USA), while Jurkat cells were cultured in RPMI-1640 (Cytvia, North Logan, UT, USA) supplemented with 10% FBS and a 1% penicillin-streptomycin solution (Beyotime, Shanghai, China). Human T cells were isolated from healthy donor peripheral blood using a Human T Cell Isolation Kit (Stemcell, Vancouver, BC, Canada) and cultured in RPMI-1640 supplemented with 10% FBS (Gibco, Clinton, OK, USA), 2 mM Glutamax (Gibco, Clinton, OK, USA), 10 mM HEPES (Gibco, Clinton, OK, USA), 55 nM β-Mercaptoethanol (ThermoFisher Scientific, Waltham, MA, USA), and 10 ng/mL IL-2 (Peprotech, Cranbury, NJ, USA).

### 2.4. Cell Growth Assay

Cancer cells in a logarithmic growth phase were evenly seeded in a 48-well plate, with 12,000–15,000 cells per well, and cultured in a 37 °C incubator. Following cell adhesion, small-molecule compounds were introduced into the culture medium, and small-molecule compounds were added to the plates during the seeding of suspension cells such as Jurkat and human T cells. Every three days, the cells were collected and quantified using CountBright™ Plus Absolute Counting Beads (Invitrogen, Hereford, TX, USA), following the manufacturer’s guidelines.

### 2.5. Cell Death Assay

HT-29 (80,000 cells/well) and LoVo (100,000 cells/well) cells were plated into 24−well plates and cultured overnight. Following this, the cells were treated with small compounds at concentrations of 20 µM and 30 µM for either 24 h or 48 h. Following treatment, the cells were harvested, rinsed with cold PBS, and subsequently stained with Annexin V FITC/PI (TransGen Biotech, Beijing, China) according to the manufacturer’s instructions. Cell death was detected by flow cytometry (ThermoFisher Scientific, Waltham, MA, USA).

### 2.6. MTT Assay

A 5 mg/mL MTT solution was prepared using MTT powder (Beyotime, Shanghai, China) and PBS. Cells (5000–7500 cells/well) were seeded into a 96-well plate. Following overnight incubation, varying concentrations of K784−8160 culture supernatants were added to each well, and the cells were further cultured for an additional 48 h and 72 h. Next, a 10% MTT solution was added to each well. After an incubation period of 1–3 h at 37 °C in the dark, the supernatant was discarded, and the formazan crystals were dissolved using DMSO. Absorbance was measured at 570 nm using a microplate reader (TECAN, Zurich, Switzerland).

### 2.7. CCK-8 Assay

HT-29 (6000 cells/well) and LoVo (7500 cells/well) cells were seeded into 96-well plates (NEST, Jiangsu, China). After incubating the cells overnight, specified concentrations of the E745−0011 and 7238−1516 culture media were introduced, followed by further culturing for another 48 h and 72 h. The original culture supernatant was then removed, and fresh medium containing 10% CCK-8 (Beyotime, Shanghai, China) was added. Subsequently, the cells were incubated in the dark for 1–4 h. Absorbance was measured at 450 nm using a microplate reader (TECAN, Zurich, Switzerland).

### 2.8. Colony Formation Assay

HT-29 and LoVo cells were plated into 12-well plates at densities of 1000 cells/well and 1500 cells/well, respectively. After overnight incubation to ensure adherence, the cells were exposed to 10, 20, or 30 μM concentrations of E745−0011 or 7238−1516, along with a DMSO control medium, and cultured at 37 °C for 10 days. During this period, the culture medium and corresponding concentrations of the compounds were replenished every 3 days. After 10 days, a 0.5% crystal violet solution (prepared with a 20% methanol solution) was used to fix colonies for 20 min. Excess crystal violet solution was rinsed off with clean water, and the plates were dried fully before photographs were captured. The number of colonies in each well was quantified using ImageJ software version 1.53 e.

### 2.9. Wound Healing Assay

Cells were seeded into 6-well plates and allowed to grow until they reached 90–100% confluence after 24 h for a wound healing assay. Wounds were generated in each well using 200 μL pipette tips. Then, the cells were treated with compounds and allowed to migrate into the wounded area. Cell migration was observed and imaged using an Inverted fluorescence microscope (4× magnification, OLYMPUS, Nagano, Japan) at different time points (0, 48 h, and 96 h). The migratory ability of the cells was analyzed using ImageJ software version 1.53 e.

### 2.10. Real-Time PCR

Total RNA was extracted from the cells using Trizol (Invitrogen, Hereford, TX, USA), following the manufacturer’s guidelines. Reverse transcription was carried out with 5 × HiScript II qRT SuperMix IIa (Vazyme, Nanjing, Chian) according to the manufacturer’s protocol. A real-time PCR was performed on a QuantStudio 3 Real-Time PCR System (ThermoFisher Scientific, Waltham, MA, USA), using 2× ChamQ SYBR qPCR Master Mix (Vazyme, China). Human TIPE3 PCR fragments were amplified using the following primers: a forward primer, 5′-TTCAGAGGGGAAAGGGACT-3′, and a reverse primer, 5′-AACATCAGGACCTGCGGC-3′. Additionally, human GAPDH PCR fragments were amplified with the following primers: a forward primer, 5′-GGAGCGAGATCCCTCCAAAAT-3′, and a reverse primer, 5′-GGCTGTTGTCATACTTCTCATGG-3′ (GeneWiz, Suzhou, China).

### 2.11. Statistical Analysis

The tools used in the computer analysis are as follows. A gnuplot was used to plot the RMSD and free energy landscape plot. USCF Chimera (USCF Chimera software version 1.16, San Francisco, CA, USA), Schrödinger (Schrödinger software version 2021-2, San Diego, CA, USA), and PyMOL (PyMOL software version 2.4, Palo Alto, CA, USA) were used to visualize and generate the 3D and 2D -protein–ligand interaction plots for complex structures from docking or MD simulations.

Experimental data were analyzed using GraphPad Prism (GraphPad Software version 9.0, Boston, MA, USA) and were presented, when appropriate, as mean ± SEM values. The statistical significance of differences between groups was determined using an unpaired Student’s *t* test, a one-way ANOVA, or a two-way ANOVA and presented as * *p* < 0.05, ** *p* < 0.01, *** *p* < 0.001, and **** *p* < 0.0001. 

## 3. Results

### 3.1. Computational Methods Aid TIPE3 Inhibitor Screening

In this work, we incorporated a DFCNN, Autodock Vina docking, DeepBindBC, MD, and a metadynamics simulation to efficiently identify potential inhibitors of TIPE3. The finally selected potential inhibitors were subjected to further experimental validation.

The screening workflow consisted of four main parts: pocket and dataset preparation, deep learning and docking-based screening, force-field-based screening, and experimental validation (Figure 1a). In the deep learning-based screening stage, the DFCNN was used initially due to its fast speed and ability to exclude non-binder compounds. Subsequently, Autodock Vina was employed to efficiently identify predicted complex structure, and the DeepBindBC was further used to evaluate the highly potential binders based on those predicted complex structures. Furthermore, Schrödinger docking was also used to explore binding conformation and stronger binders. By applying score cutoff values of 0.99, 0.99, −9.5 Kcal/mol, and −8.8 Kcal/mol for the DFCNN, DeepBindBC, Autodock Vina, and Schrödinger docking, the initial candidate pool was narrowed down to 58 compounds.

In the subsequent screening stage, a pocket MD simulation and metadynamics simulation were utilized. Finally, six compounds were selected for experimental testing; their chemical structures are listed (Figure 1b). All the scores for the six compounds are shown in Table 1. All six compounds exhibited very favorable binding scores. Among them, ZINC000009238243 (7238-1516) demonstrated the lowest docking score in both VINA and Schrödinger docking and was later validated to have active anti-cancer properties. However, the results obtained via the VINA and Schrödinger docking methods were not always consistent, suggesting that the use of both methods could be beneficial. For example, while ZINC000011215475 showed the second lowest score in VINA docking, it exhibited the highest score in Schrödinger docking. Conversely, ZINC000021790483 demonstrated the highest score in VINA docking but the second-lowest score in Schrödinger docking. Interestingly, neither ZINC000011215475 nor ZINC000021790483 showed anti-cancer activity in subsequent experimental validations. We further checked the RMSD values for the six compounds during the 40 ns MD simulation, which also indicated that most of the ligands exhibited relatively low and stable deviations from their initial conformations (Figure 2a). To ensure full equilibration, we also observed the analysis result of only the last 20 ns (30~40 ns) to check the binding stability. We selected these six compounds mainly based on free energy landscapes predicated using a metadynamics simulation (Figure 2b). We assigned a value of 1 for the MD simulation if the trajectory RMSD was relatively small and exhibited less fluctuation throughout the MD simulation. We assigned a value of 1 for the metadynamics simulation if the lowest energy basin of the calculated free energy landscape was much larger than 0. It is important to acknowledge that these scores were predominantly assigned based on empirical judgment rather than precise calculations. It clearly revealed that all six selected compounds exhibited their lowest energy basin at a coordination number significantly greater than 0. This suggests that the ligands maintained structural stability throughout the simulation and indicates favorable binding behavior for these compounds.

### 3.2. Experimental Validation for the Anti-Tumor Effects of Candidate Inhibitors Targeting TIPE3

TIPE3 expression levels vary greatly in different human cell lines [4,5]. We also examined the relative mRNA levels of TIPE3 in several human cell lines (Appendix A). To explore the anti-tumor effects of the six candidate compounds mentioned above, we selected human colon cancer cell lines HT-29 and LoVo (with high TIPE3 expression), the human esophageal carcinoma cell line TE-1 (with moderate TIPE3 expression), and the human T lymphocyte cell line Jurkat (with little or no TIPE3 expression) for further experimental studies.

Firstly, we performed a CCK-8 assay and MTT assay to investigate the cytotoxic activities of the six selected small-molecule compounds against HT-29, LoVo, and TE-1. Three small molecules, K784-8160, E745-0011, and 7238-1516, exhibited potent anti-tumor activities. As shown in Figure 3a, K784-8160 induced a dose-dependent decrease in the viability of HT-29 cells and LoVo cells, with IC50 values of 4.94 μM and 26.35 μM, respectively. Treatment with E745-0011 and 7238-1516 also significantly reduced the viability of HT-29 cells and LoVo cells in a concentration-dependent manner (Figure 3b,c). The IC50 values of E745-0011 and 7238-1516 in the LoVo cells were determined to be 16.38 μM and 14.02 μM, respectively, using GraphPad Prism 9.0 software. However, the other three small molecules (STL125869, 7114090881, and D393-0320) showed little or no killing activity against cancer cells below a concentration of 20 μM (Figure 3d–f). Therefore, we focused on K784-8160, E745-0011, and 7238-1516 for further study.

To further validate the effects of K784-8160 on cancer cells, we performed flow cytometry to enumerate the number of -colon adenocarcinoma cells (HT-29 and LoVo) expressing TIPE3 highly in the culture. As shown in Figure 4a, K784-8160 significantly reduced the number of cancer cells at a concentration of 20 μM. Consistent with this finding, colony formation by HT-29 and LoVo cells was markedly inhibited by K784-8160 (Figure 4b,c). Treatment with K784-8160 also significantly decreased the migration of HT-29 and LoVo cells in the wound healing assay compared to the DMSO control (Figure 4d,e). These results indicate that K784-8160 may have potent inhibitory activities against TIPE3-expressing tumor cells.

To establish whether the effect of K784-8160 is related to TIPE3, we added it to the cultures of human Jurkat cells and normal human T lymphocytes, which expressed no detectable TIPE3 (Appendix A). The viability of both the Jurkat cells and human primary T cells was significantly reduced by K784-8160 (Figure 4f–h). Thus, K784-8160 may possess non-selective toxicity toward tumor cells and normal human cells.

To further confirm the activities of E745-0011 and 7238-1516 against cancer cells, we next conducted a cell growth assay on HT-29, LoVo, and Jurkat cells treated with 20 μM E745-0011, 7238-1516, or a control medium for 9 days. Both compounds significantly reduced the number of HT-29 and LoVo cells (Figure 5a and Figure 6a). However, neither E745-0011 nor 7238-1516 exhibited a significant effect on the growth of Jurkat cells compared to the DMSO control (Figure 5a and Figure 6a). Moreover, treatment with E745-0011 and 7238-1516 significantly decreased the colony formation of HT-29 and LoVo cells (Figure 5b,c and Figure 6b,c). To assess the effect of these compounds on cell death, the apoptosis of HT-29 and LoVo cells was analyzed after treatment with E745-0011 and 7238-1516. The results showed that total number of apoptotic cells in the 20 μM treatment group was nearly doubled compared to the DMSO control group (Figure 5d and Figure 6d). We then investigated whether E745-0011 and 7238-1516 exhibited any cytotoxic effects on normal human cells. Human primary T cells were treated with various concentrations of E745-0011 or 7238-1516 for 72 h, and cell viability was assessed using the CCK-8 assay. The results revealed that neither compound significantly affected the viability of human T cells at concentrations below 30 μM (Figure 5e and Figure 6e). Furthermore, we treated human T cells separately with 20 μM of E745-0011 or 7238-1516 for 9 days and monitored cell growth every 3 days. The data showed that neither E745-0011 nor 7238-1516 had a substantial effect on the total number of human T cells compared to the DMSO control (Figure 5f and Figure 6f). These findings suggest that E745-0011 and 7238-1516 may specifically impact the viability, growth, proliferation, and apoptosis of TIPE3-expressing cancer cells while showing minimal cytotoxicity toward normal human T cells.

### 3.3. Molecular Docking for Inhibitors and TIPE3

We analyzed the interactions of the docked structures of the three found active compounds and TIPE3 in detail, shown in Figure 7. The primary interactions between 7238-1516 and TIPE3 are hydrophobic and Pi-related interactions in nature, as shown in Figure 7a. Specifically, there are several LEU, PHE, and ILE residues that are predominantly involved in hydrophobic interactions with 7238-1516. Similarly, the E745-0011 and TIPE3 pocket predominantly engages in hydrophobic and electrostatic interactions, as shown in Figure 7b. Specifically, the methyl benzene ring of E745-0011 establish π-π stacking interactions with residues of PHE67. In the interaction of K784-8160 with the TIPE3 binding site, presented in Figure 7c, a multitude of pocket residues are implicated. Notably, LEU (71, 74, 78, and 19) PHE (127, 16), ILE15, and MET6 predominantly form hydrophobic contacts with K784-8160. Furthermore, PHE127 forms significant π-π stacking interactions with the benzene-like ring of K784-8160. Additionally, THR74 and SER12 demonstrate polar interactions with K784-8160.

In summary, all three active compounds extensively engage in hydrophobic interactions with a number of hydrophobic residues including ILE, PHE, and ILE. Moreover, these compounds share several common interaction residues, as indicated by their 2D interaction plots. Residues such as LEU71, PHE16, and PHE127 are notably involved in important interactions with all three compounds. Furthermore, we notice that 7238-1516 and E745-0011 have more common interacting residues, such as ILE47, PHE67, and LEU40, which may explain their lesser effect on human normal cells. The presence of these shared interacting residues suggests their critical role in facilitating the successful docking of ligands within the binding pocket. Additionally, each of the three compounds resembles a hand with three fingers such that the palm is adorned with polar atoms and the fingers are structured like benzene rings. All of this provides us with abundant clues to design more promising inhibitors.

## 4. Discussion

TIPE3 is highly upregulated in several human cancers and has been established to play an important role in tumor progression. Whether inhibiting the function of TIPE3 with small molecules could be an effective strategy for cancer treatment is unknown. Here, we report three potential small-molecule inhibitors of TIPE3 with potent in vitro anti-tumor activities. Despite the promising findings, several challenges merit attention. While E745-0011 and 7238-1516 display potent anti-tumor activities, further studies delineating their mechanisms of action and in vivo efficacy are crucial for evaluating their therapeutic potential comprehensively. Improving the accuracy and predictive capacity of computational models is imperative to refine inhibitor screening, enabling the identification of candidates with high specificity and efficacy while reducing potential off-target effects. Integrating additional structural and physicochemical parameters can enhance precision in identifying lead compounds with improved pharmacological profiles. Furthermore, limitations associated with acquiring existing TIPE3 protein and reliable antibodies hampered direct drug and TIPE3 affinity testing. Obtaining high-quality recombinant TIPE3 proteins and specific antibodies is pivotal for precise molecular studies and inhibitor validation. Efforts directed toward developing standardized TIPE3 proteins and antibodies, along with reliable assays for compound—TIPE3 interaction studies, are critical for advancing TIPE3 inhibitor research. Collaboration and focused research within the scientific community are crucial to overcoming these fundamental challenges in TIPE3-related studies.

In our previous research, we carried out a screening process on TIPE2 and identified a ligand (UM-164) that demonstrated significant binding affinity, as detailed in our earlier publication [34]. It is important to note the remarkable similarity between the binding sites of TIPE2 and TIPE3—both are characterized by spacious protein cavities. The affinity of the ligands discovered during this screening process suggests a high likelihood that their anti-cancer effects are mediated through interactions with these cavities.

Building on these findings, the TIPE3- compound complexes predicted in this study are highly valuable for guiding subsequent rounds of drug optimization. For example, examining the compounds E745-0011 and 7238-1516 revealed minimal contact with the bottleneck residue of the cavity. This observation becomes particularly relevant when considering the charges and the presence of aromatic residues within this bottleneck region. An extension of the compounds to establish direct interactions with these residues could potentially enhance binding affinity, aiding in the development of novel drug candidates.

During the later phases of the screening, metadynamics were employed to calculate the free energy landscapes to select potential candidates. While this method provided a useful approximation of binding preferences, which is adequate for screening purposes, it is acknowledged that standard metadynamics may not always offer an accurate determination of binding affinity. This limitation was addressed by suggesting the application of more refined computational methods, such as funnel metadynamics, for the precise quantification of binding affinities and supporting the design of high-affinity drugs.

Our screening encompassed the ZINC database (version 2019), and this methodology could be applied to subsequent versions of ZINC or other popular databases like ChemDiv or the Natural Product Database. Moreover, combining our screening with cutting-edge, deep learning-based compound generation models—such as DrugGPT and LSTM Chem—not only enhances the ability to identify de novo compounds but also reflects the adaptability of our pipeline. Our screening process can be customized by integrating additional steps tailored to the user’s specific requirements, including the implementation of our newly developed DeepBindGCN_RG/BC algorithms.

The ChemGPT-4.7M model (https://huggingface.co/ncfrey/ChemGPT-4.7M, accessed on 20 July 2023) is capable of generating valid chemical compound structures. Considering that the drug-like chemical space is vast, ranging from 10^30^ to 10^60^ possibilities, the subset of compounds with anticancer properties is likely much smaller and characterized by unique properties. To refine the breadth of the screening dataset while ensuring high-quality candidates, we propose a fine-tuning approach using a known cancer compound dataset. This will generate a model specifically trained to produce potential anti-cancer compounds, facilitating the creation of a targeted screening database for cancer drug discovery. After the fine-tuning process, we developed a new model named ChemGPT-cancer, which can be accessed at https://github.com/haiping1010/ChemGPT-cancer (accessed on 10 March 2024). Integrating our screening pipeline with ChemGPT-cancer provides a feasible pathway toward expediting the discovery of active anti-cancer compounds and substantially augmenting the existing repertoire of anti-cancer drugs. Our team also generated an initial compound library using ChemGPT-cancer and conducted a preliminary screening targeting the TIPE3 receptor. While these initial screening results await experimental validation and are not extensively elaborated upon in this manuscript, we believe that sharing this screening protocol and the compound library generated is crucial for fostering future research in similar technology applications. Hence, we have made the relevant software and datasets available on GitHub, aiming to support and encourage other research teams to explore and harness such tools for innovative anti-cancer drug-screening endeavors. This open resource-sharing approach, coupled with our preliminary efforts in this field, sets a foundation for collaborative advancement in cancer drug discovery.

## 5. Conclusions

In summary, utilizing comprehensive computational screening, we effectively filtered out six potential TIPE3 inhibitors for experimental validation. Among them, three small-molecule compounds (K784-8160, E745-0011, and 7238-1516) showed potent anti-tumor activities. K784-8160 exhibited anti-tumor effects on various cancer cell lines, albeit showing toxicity toward normal human cells. Conversely, E745-0011 and 7238-1516 demonstrated significant cancer cell suppression with minimal cytotoxicity towards healthy cells, indicating their potential as targeted anti-cancer agents. A molecular docking analysis revealed their engagement in crucial hydrophobic interactions with TIPE3, portraying potential binding mechanisms. Shared interaction residues offer insights for designing potent inhibitors against TIPE3-associated cancers. These findings not only provide a promising avenue for the development of targeted therapies against TIPE3 in cancer treatment but also establish a model for harnessing sophisticated computational strategies to streamline the discovery and optimization of new inhibitors.

## Figures and Tables

**Figure 1 cells-13-00771-f001:**
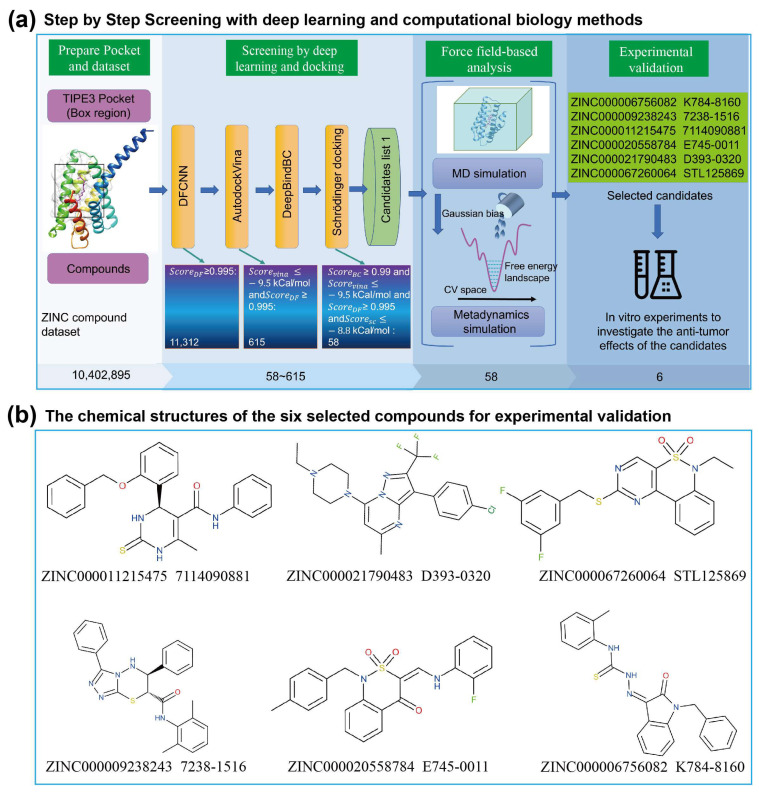
The virtual screening procedure integrates deep learning, docking, and force-field-based methods to identify highly reliable drug candidates for TIPE3. (**a**) A schematic diagram of the screening of TIPE3 inhibitors from the ZINC dataset. (**b**) The chemical structures of the six compounds selected for experimental validation.

**Figure 2 cells-13-00771-f002:**
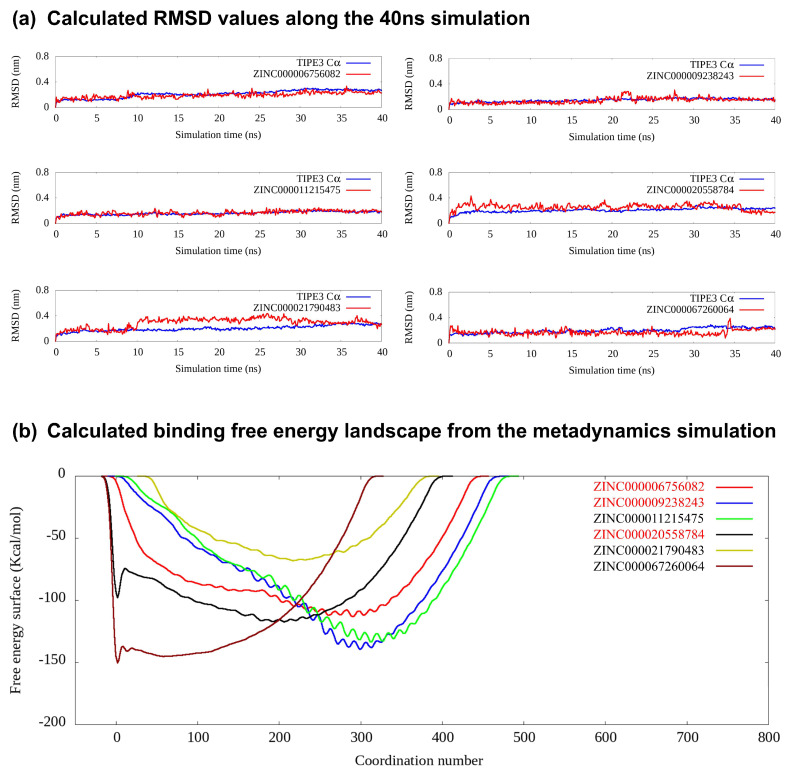
The MD- and metadynamics simulation-related analysis. (**a**) RMSD values during a 40 ns simulation for the six selected inhibitors of TIPE3. (**b**) The calculated binding free energy landscape from the metadynamics simulation of the six selected compounds; the three labeled in red are the experimentally validated active compounds.

**Figure 3 cells-13-00771-f003:**
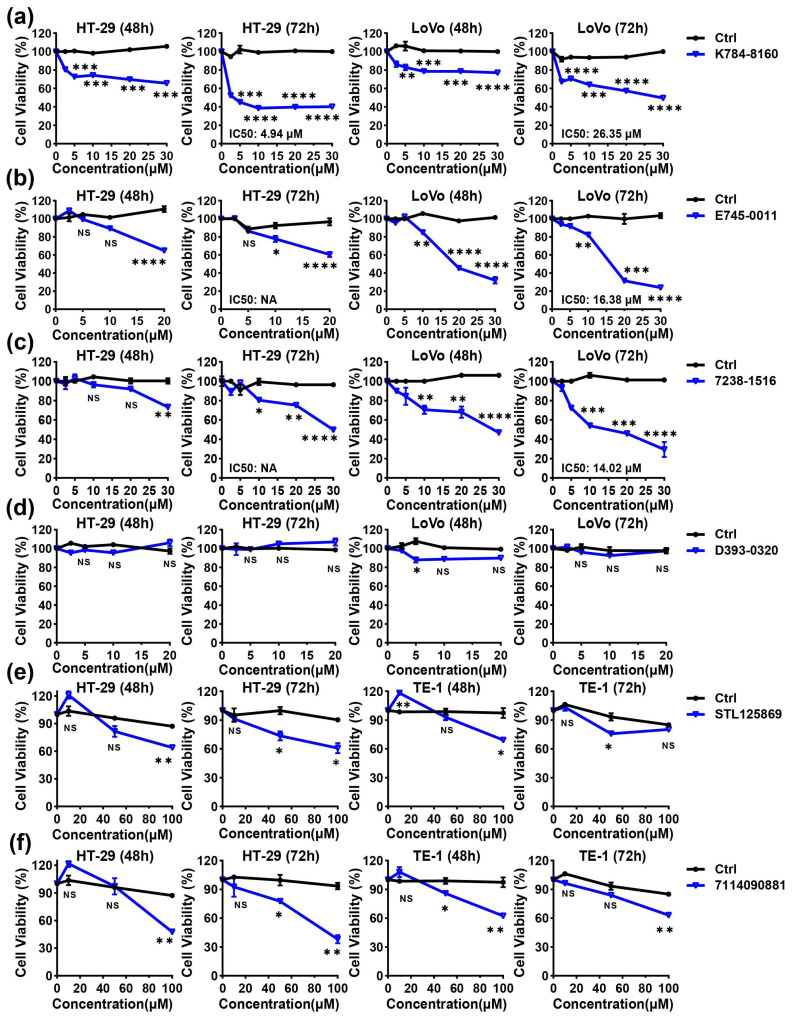
Cytotoxic activity of the six selected small-molecule compounds on cancer cell lines in vitro. (**a**–**d**) HT-29 and LoVo cells were treated with different concentrations of K784-8160, E745-0011, 7238-1516, D393-0320, or a DMSO control for 48 h or 72 h. Cell viability was determined using a CCK-8 assay. (**e**,**f**) HT-29 and TE-1 cells were treated with different concentrations of STL125869, 711409088, or a DMSO control for 48 h or 72 h. Cell viability was determined using an MTT assay. To exclude the effect of DMSO on cell viability, we set up a DMSO control group for each indicated drug concentration (by adding an equal volume of DMSO as in the experimental group). Data were expressed as mean ± SEM values; *n* = 3. NS, not significant (*p* > 0.05); * *p* < 0.05, ** *p* < 0.01, *** *p* < 0.001, and **** *p* < 0.0001. Significance was determined using an unpaired Student’s *t* test. The experiments were performed at least three times with similar results.

**Figure 4 cells-13-00771-f004:**
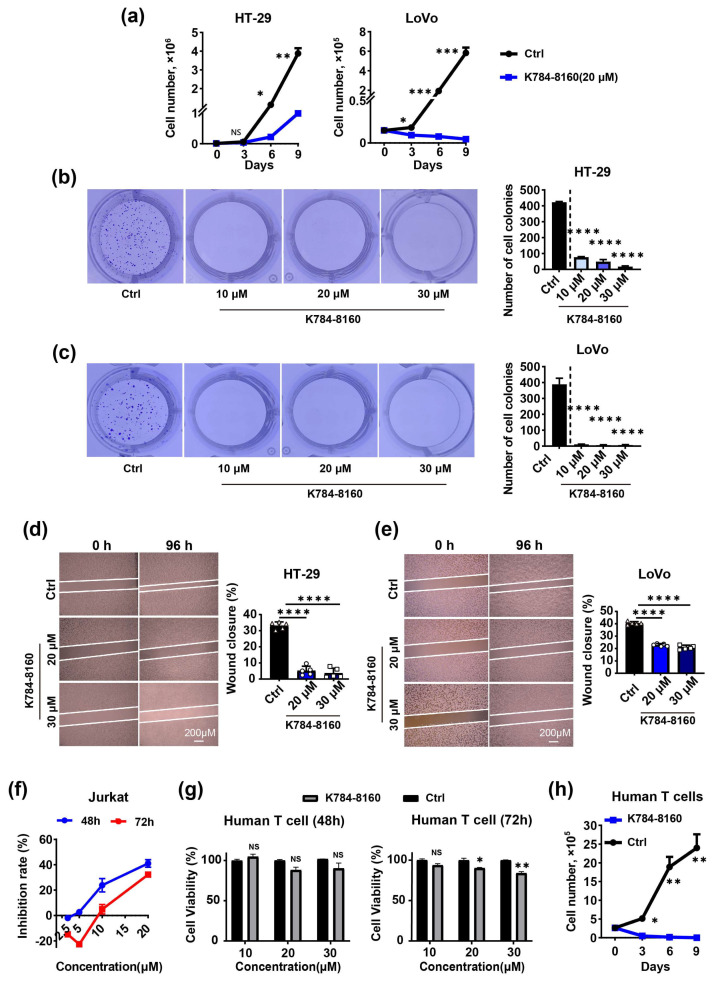
K784-8160 exhibits non-selective toxicity toward both tumor cells and normal human cells. (**a**) A cell growth analysis of human HT-29 and LoVo cells after either treatment with K784-8160 at a concentration of 20 μM or a DMSO control over the indicated times. (**b**,**c**) Numbers of colonies of HT-29 and LoVo cells after treatment with different concentrations of K784-8160 were detected by a colony formation assay. (**d**,**e**) The migration of HT-29 and LoVo treated with or without K784-8160 was determined using a wound healing assay. The wound closure rates of HT-29 and LoVo cells treated with or without K784-8160 for 96 h were calculated separately. The corresponding bar chart is shown on the right. (**f**) Jurkat cells were treated with K784-8160 at different doses for 48 h and 72 h. Cell viability was assessed by a CCK-8 assay. (**g**) Human T cells were treated with different doses of K784-8160 for 48 h and 72 h. Cell viability was determined using a CCK-8 assay. (**h**) A cell growth analysis of human T cells after either treatment with K784-8160 at a concentration of 20 μM or a DMSO control over the indicated times. Data were expressed as mean ± SEM values. NS, not significant (*p* > 0.05); * *p* < 0.05, ** *p* < 0.01, *** *p* < 0.001, and **** *p* < 0.0001. Significance was determined using an unpaired Student’s *t* test. The experiments were performed at least three times with similar results.

**Figure 5 cells-13-00771-f005:**
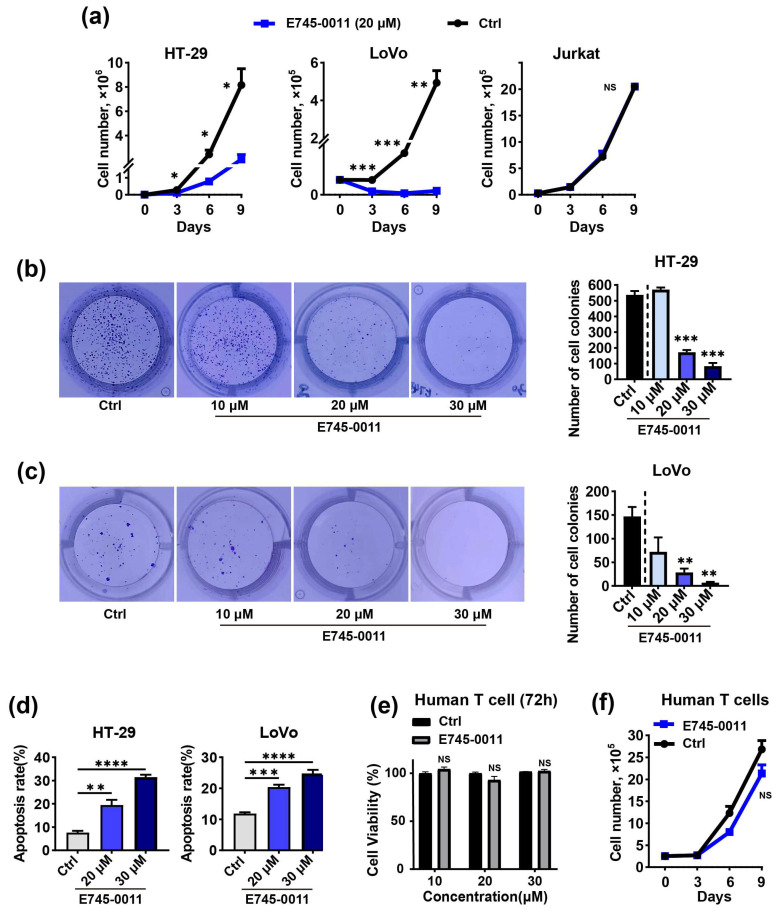
E745-0011 is effective against colon-cancer-derived cell lines in vitro while having no toxicity to normal human primary cells. (**a**) A cell growth analysis of HT-29, LoVo, and Jurkat cells after treatment with either 20 μΜ E745-0011 or a DMSO control over the indicated times. (**b**,**c**) Numbers of colonies of HT-29 and LoVo cells treated with different concentrations of E745-0011, as determined by a colony formation assay. (**d**) HT-29 and LoVo cells were treated with different concentrations of E745-0011 for 48 h. Apoptosis was measured by annexin V/propidium iodide staining and flow cytometry. (**e**) Human primary T cells were treated with different concentrations of E745-0011 for 72 h, and cell viability was determined using a CCK-8 assay. (**f**) A cell growth analysis of human T cells after treatment with either 20 μΜ E745-0011 or a DMSO control over the indicated times. Data were expressed as mean ± SEM values; *n* = 3. NS, not significant (*p* > 0.05); * *p* < 0.05, ** *p* < 0.01, *** *p* < 0.001, and **** *p* < 0.0001. In (**a**–**c**,**e**–**f**), significance was determined using an unpaired Student’s *t* test. Significance in (**d**) was determined by a one-way ANOVA. The experiments were performed at least three times with similar results.

**Figure 6 cells-13-00771-f006:**
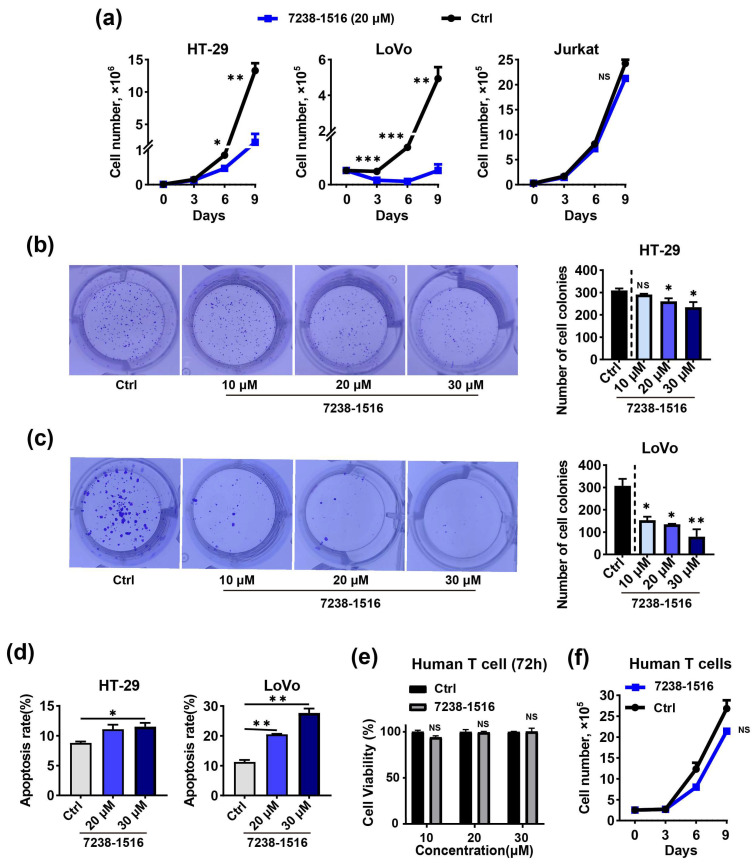
7238-1516 is effective against colon-cancer-derived cell lines in vitro while having no toxicity to normal human primary cells. (**a**) A cell growth analysis of HT-29, LoVo, and Jurkat cells after treatment with either 20 μΜ 7238-1516 or a DMSO control over the indicated times. (**b**,**c**) Numbers of colonies of HT-29 and LoVo cells treated with different concentrations of 7238-1516, as determined by a colony formation assay. (**d**) HT-29 and LoVo cells were treated with 7238-1516 for 48 h and 24 h, respectively. Apoptosis was measured by flow cytometry. (**e**) Human T cells were treated with different concentrations of 7238-1516 for 72 h, and cell viability was determined using a CCK-8 assay. (**f**) A cell growth analysis of human T cells after treatment with either 20 μΜ 7238-1516 or a DMSO control over the indicated times. Data were expressed as mean ± SEM values; *n* = 3. NS, not significant (*p* > 0.05); * *p* < 0.05, ** *p* < 0.01, *** *p* < 0.001. In (**a**–**c**,**e**–**f**), significance was determined using an unpaired Student’s *t* test. Significance in (**d**) was determined by a one-way ANOVA. The experiments were performed at least three times with similar results.

**Figure 7 cells-13-00771-f007:**
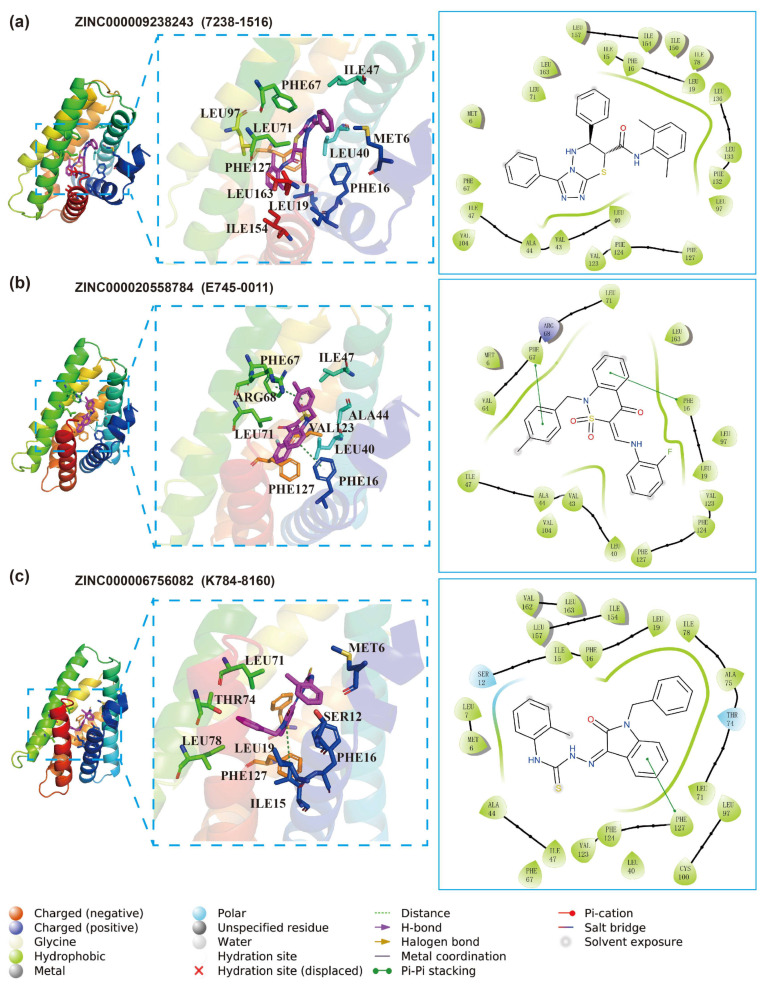
Interaction analysis of TIPE3 with compounds identified from this study with docked conformations. (**a**) Compound ZINC000009238243 (7238-1516) binding to the TIPE3, displaying both 3D detailed interactions (middle) and 2D representation (right). (**b**) Interaction between ZINC000020558784 (E745-0011) and TIPE3. (**c**) Interaction between ZINC000006756082 (K784-8160) and TIPE3. Residues in all 3D visualizations were color-coded by rainbow in PyMOL. The 2D diagrams employ colors and symbols as standardized by the Schrödinger 2D interaction plots.

**Table 1 cells-13-00771-t001:** The prediction scores of different methods for the 6 selected compounds. Notably, we used score 1 for stable binding during MD and utilized score 1 to represent the ligand’s preference for binding to the protein according to the free energy landscape calculated via metadynamics.

ZINC ID	DFCNN	VINA Docking (Kcal/mol)	DeepBindBC	Schrödinger Docking (Kcal/mol)	MD Simulation	Metadynamics
ZINC000009238243	1.00	−10.9	0.99	−9.39	1	1
ZINC000021790483	1.00	−9.6	0.99	−9.23	1	1
ZINC000067260064	1.00	−9.9	0.99	−8.89	1	1
ZINC000006756082	1.00	−9.9	1.00	−8.87	1	1
ZINC000020558784	1.00	−9.8	1.00	−8.86	1	1
ZINC000011215475	1.00	−10.1	1.00	−8.82	1	1

## Data Availability

The data presented in this study are available upon request from the corresponding author.

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
