# Peer review of "Small-Molecule Inhibitors of TIPE3 Protein Identified through Deep Learning Suppress Cancer Cell Growth In Vitro"

_cells, 2024, doi:10.3390/cells13090771_

Round 1

Reviewer 1 Report

Comments and Suggestions for Authors

This study investigates the potential of targeting TIPE3, a protein associated with cancer cell growth, for therapeutic purposes. Employing sophisticated AI techniques, such as DFCNN and a variety of docking simulations, the research identified six potential inhibitors of TIPE3 from the ZINC compound dataset. It was discovered that three of these compounds exhibit significant anti-tumor activity in vitro, with two demonstrating selective cytotoxicity towards cancer cells that overexpress TIPE3. The manuscript is interesting and well-structured; however, I have a few suggestions prior to publication.

1.     The use of molecular docking to validate their interactions with TIPE3 is noteworthy, but the discussion on the trends observed with VIVA and Schrodinger docking could be elaborated further, particularly on page 284.

2.     The stability of the initial conformations in the simulation results presented in Figure 2 raises questions. Is there a period required for stabilization? Has the optimization of the binding position been considered?

3.     The choice of the four cell lines, HT29, LoVo, TE-1, HCT116, and Jurkat, warrants clarification. Specifically, for the colon adenocarcinoma cells (HT29 and LoVo) discussed on page 328, additional references could be beneficial. This applies equally to the other two cell lines.

4.     The text in Figures 4d and 4e is excessively small and needs enlargement for better readability.

Author Response

We would like to thank the reviewers for their insightful comments on our manuscript.  In this revised manuscript, we have made every effort to address the concerns raised by the reviewers. Below are our point-by-point responses to the reviewers’ comments (shown in blue fonts).

Reviewer #1:

This study investigates the potential of targeting TIPE3, a protein associated with cancer cell growth, for therapeutic purposes. Employing sophisticated AI techniques, such as DFCNN and a variety of docking simulations, the research identified six potential inhibitors of TIPE3 from the ZINC compound dataset. It was discovered that three of these compounds exhibit significant anti-tumor activity in vitro, with two demonstrating selective cytotoxicity towards cancer cells that overexpress TIPE3. The manuscript is interesting and well-structured; however, I have a few suggestions prior to publication.

  1. The use of molecular docking to validate their interactions with TIPE3 is noteworthy, but the discussion on the trends observed with VIVA and Schrodinger docking could be elaborated further, particularly on page 284.

Response: We thank the reviewer for raising these important questions, we now added the following discussion on the trends observed with autodock Vina and Schrodinger docking in the revised manuscript (Pages 6-7, Lines 300-310):

“All the scores for the 6 compounds are shown in Table 1. All six compounds exhibited very favorable binding scores. Among them, ZINC000009238243(7238-1516) demon-strated the lowest docking score in both VINA and Schrödinger docking, which was later validated as active in anti-cancer properties. However, the results from VINA and Schrödinger docking methods were not always consistent, suggesting that the use of both methods could be beneficial. For example, while ZINC000011215475 showed the second lowest score in VINA docking, it exhibited the highest score in Schrödinger docking. Conversely, ZINC000021790483 demonstrated the highest score in VINA docking, but the second lowest score in Schrödinger docking. Interestingly, neither ZINC000011215475 nor ZINC000021790483 showed anti-cancer activity in subsequent experimental validation.”

  1. The stability of the initial conformations in the simulation results presented in Figure 2 raises questions. Is there a period required for stabilization? Has the optimization of the binding position been considered?

Response: Yes, this is a very good point. We have a 0.1ns NVT and 10ns NPT for equilibration (stabilization) before carrying out the 40 ns production MD simulation. We have used the last frame of 40 ns MD simulation as a good binding position for later analysis, which is a common practice. We use the best Schrödinger docking conformation (corresponding to best score) as the initial conformation of MD simulation. To ensure the full equilibration, we also observe the analysis result of only the last 20ns (30~40ns) to check the binding stability. We have added this detailed description to “Supplementary material”.

  1. The choice of the four cell lines, HT-29, LoVo, TE-1, HCT 116, and Jurkat, warrants clarification. Specifically, for the colon adenocarcinoma cells (HT29 and LoVo) discussed on page 328, additional references could be beneficial. This applies equally to the other two cell lines.

Response: We thank the reviewer for the above suggestions. The selection of these cell lines (including HT-29, LoVo, TE-1, HCT 116, and Jurkat) was made after careful consideration. We first searched the literature and analyzed the expression levels of TIPE3 in different tumor cell lines using public databases. Then, we verified the mRNA levels of TIPE3 in tumor cell lines available in our laboratory by RT-PCR (data shown in Supplementary Figure 1). To assess the anti-tumor activity and specificity of the selected potential inhibitors of TIPE3, we ultimately chose human colon cancer cell line HT-29 and LoVo (with high TIPE3 expression), human esophageal carcinoma cell line TE-1 (with moderate TIPE3 expression), and human T lymphocyte cell line Jurkat (with little or no detectable TIPE3 expression) for further experimental studies. Accordingly, we have added relevant supporting references (Page 9, Line 341) and supplementary explanations to Page 9, Lines 341-343.

  1. The text in Figures 4d and 4e is excessively small and needs enlargement for better readability.

Response: We thank the reviewer for the suggestion. The text in Figures 4d and 4e has been revised to improve the readability (Page 13).

Reviewer 2 Report

Comments and Suggestions for Authors

The manuscript by Chen et al. describes how new inhibitors of the TIPE3 protein were identified using theoretical methods derived from artificial intelligence and molecular mechanics. Some of the small molecules discovered by the authors were tested in vitro.

The results are interesting, although there hasn't been any direct experimental observation of the interaction between the TIPE3 protein and the small molecule found by the authors. 

In my opinion, there are only a few points that need to be corrected by the authors before publication:

- Lines 100 and 105. The reference to the article cited by the authors is missing.

- The units used throughout the paper should be the same, for example: distance in lines 122 and 139, energy in Figure 2.b and Table 1.

-Table 1. How were the scores 1 calculated for MD and metadynamics simulations?

- Line 150. The authors should describe in detail how the molecular dynamics and metadynamics simulations were performed. There's no "Supplementary Section 1"

- There are two figures S1 ("S" stands for supplementary?), line 304 and line 352. The authors should check which figure is in the manuscript and which is in the supplementary material and adjust the text accordingly.

- Figure 2. The axis limits chosen by the authors make it difficult to see the differences between the graphs, I would suggest the following changes:

*Figure 2.a The maximum limits of the RMSD axis should be lowered from 1.6 nm to 0.5-0.7 nm. 

*Figure 2.b The minima of the free energy surface should be increased from -1600 kJ/mol to -900 or -1000 kJ/mol. 

Author Response

We would like to thank the reviewers for their insightful comments on our manuscript.  In this revised manuscript, we have made every effort to address the concerns raised by the reviewers. Below are our point-by-point responses to the reviewers’ comments (shown in blue fonts).

Reviewer #2:

The manuscript by Chen et al. describes how new inhibitors of the TIPE3 protein were identified using theoretical methods derived from artificial intelligence and molecular mechanics. Some of the small molecules discovered by the authors were tested in vitro.

The results are interesting, although there hasn't been any direct experimental observation of the interaction between the TIPE3 protein and the small molecule found by the authors.

In my opinion, there are only a few points that need to be corrected by the authors before publication:

  1. Lines 100 and 105. The reference to the article cited by the authors is missing.

Response: We apologize for missing the references in the original manuscript. We have added the relevant references to the revised manuscript (Pages 3 and 4, Lines 115-177).

  1. H. Zhang, L. Liao, Y. Cai, Y. Hu, and H. Wang, “IVS2vec: A tool of Inverse Virtual Screening based on word2vec and deep learning techniques,” Methods, vol. 166, pp. 57–65, Aug. 2019.
  2. H. Zhang, T. Zhang, K. M. Saravanan, L. Liao, H. Wu, H. Zhang, H. Zhang, Y. Pan, X. Wu, and Y. Wei, “DeepBindBC: A practical deep learning method for identifying native-like protein-ligand complexes in virtual screening,” Methods, vol. 205, pp. 247–262, 2022.
  3. H. Zhang, K. M. Saravanan, Y. Yang, M. T. Hossain, J. Li, X. Ren, Y. Pan, and Y. Wei, “Deep Learning Based Drug Screening for Novel Coronavirus 2019-nCov.,” Interdisciplinary sciences, computational life sciences, vol. 12, no. 3, pp. 368–376, Sep. 2020.
  4. H. Zhang, Y. Yang, J. Li, M. Wang, K. M. Saravanan, J. Wei, J. Tze-Yang Ng, M. Tofazzal Hossain, M. Liu, H. Zhang, X. Ren, Y. Pan, Y. Peng, Y. Shi, X. Wan, Y. Liu, and Y. Wei, “A novel virtual screening procedure identifies Pralatrexate as inhibitor of SARS-CoV-2 RdRp and it reduces viral replication in vitro.,” PLoS computational biology, vol. 16, no. 12, p. e1008489, 2020.
  5. H. Zhang, J. Li, K. M. Saravanan, H. Wu, Z. Wang, D. Wu, Y. Wei, Z. Lu, Y. H. Chen, X. Wan, and Y. Pan, “An Integrated Deep Learning and Molecular Dynamics Simulation-Based Screening Pipeline Identifies Inhibitors of a New Cancer Drug Target TIPE2,” Frontiers in Pharmacology, vol. 12, p. 3297, 2021.
  6. S. Jaeger, S. Fulle, and S. Turk, “Mol2vec: Unsupervised Machine Learning Approach with Chemical Intuition,” Journal of Chemical Information and Modeling, vol. 58, no. 1, pp. 27–35, Jan. 2018.

  1. The units used throughout the paper should be the same, for example: distance in lines 122 and 139, energy in Figure 2.b and Table 1.

Response: Thanks for the helpful suggestion; we have used same unit throughout the revised manuscript.

  1. Table 1. How were the scores 1 calculated for MD and metadynamics simulations?

Response: Thanks for the helpful suggestion. We assign 1 for the MD simulation if the trajectory RMSD is relatively small and less fluctuation throughout the MD simulation. We assign 1 for the metadynamics simulation if the lowest energy basin of calculated free energy landscape is much larger than 0. It is important to acknowledge that these scores are predominantly assigned based on empirical judgment rather than precise calculations. We have added relevant explanation to the revised manuscript (Page 7, Lines 316-323).

  1. Line 150. The authors should describe in detail how the molecular dynamics and metadynamics simulations were performed. There's no "Supplementary Section 1"

Response: We apologize for not including “Supplementary Section 1” in the supplementary materials. We have changed the file name, replacing “Supplementary Section 1” with “Supplementary material” (Page 4, Line 179). In this supplementary material, we described how the molecular dynamics and metadynamics simulations were performed in detail. We will upload the “Supplement material” in this revised submission.

  1. There are two figures S1 ("S" stands for supplementary?), line 304 and line 352. The authors should check which figure is in the manuscript and which is in the supplementary material and adjust the text accordingly.

Response: We thank the reviewer for raising these issues. We apologize for the oversight and inaccuracies. For better understanding, we have revised “Figure S1” to “Supplementary Figure 1” (Page 10, Lines 349 and 351). Additionally, we have inserted Supplementary Figure 2 as a part of Figure 4 as suggested by Reviewer 3 (Page 13).

  1. Figure 2. The axis limits chosen by the authors make it difficult to see the differences between the graphs, I would suggest the following changes:

*Figure 2.a The maximum limits of the RMSD axis should be lowered from 1.6 nm to 0.5-0.7 nm.

*Figure 2.b The minima of the free energy surface should be increased from -1600 kJ/mol to -900 or -1000 kJ/mol.

Response: Thanks for these helpful suggestions. The maximum limits of the RMSD axis of Figure 2.a have been lowered from 1.6 nm to 0.8 nm (0.7nm will lead to slightly overlap of label and curve for few cases) and the minima of the free energy surface of Figure 2.b has been increased from about -400 Kcal/mol to -250 Kcal/mol now. The revised Figure 2 is much easier to see the differences between the graphs (Page 9).

Reviewer 3 Report

Comments and Suggestions for Authors

The manuscript, devoted to the computer calculation of ligands for intracellular targets, represents a modern direction in the use of high-performance computing to search for ligands for receptors and enzymes. Readers of biological journals are more interested in practical results and reviews that show the influence of a molecule on another molecule, a system, or an organism as a whole. The authors tried to present a comprehensive study, but the good computational part was not fully supported by biological experiments.

Part 3.4. De novo screening... is completely unsupported by experiments and should be removed. If the authors consider the results of this part interesting, then this is a reason to publish another article. This article takes these unsubstantiated predictions out of context.

Problems exist in the presentation of the material, or rather in the repeated repetition of the same facts in different sections without developing any ideas and comparing with previously published data. I propose to completely remove the first two paragraphs of the Discussion and whole Conclusion. The purpose of the article is not to write down facts several times for memorization, but to briefly convey important results.

On page 12 Figure S1. K784-8160 exhibits... this should be a second supplementary figure not S1, but I would to recommend to insert it as a part of Figure 4. Also, Figure S1 on page 9 as important for understand should be included in the main text as well. In Figure 3, the graphs show the data as viability to a concentration in micromoles. This is true for molecules studied, but the graphs show DMSO controls that do not have concentration dependence. My question is why the values for DMSO are flexible, perhaps it was a statistical gap. Are data normalized?  Do authors used mathematical statistic methods for comparison.

Looks like the data collection wasn't very good. For many tests, n=3, whereas the optimal recommended replicates are 5. In Methods:

2.6. MTT analysis

2.7. Cell viability assay

2.8. Colony formation assay

The repetition of the experiment is indicated as double.

This is not indicated for other tests, but it seems to me that they were also done with low repeatability.

It is necessary to redo experiments with 3-5 repetitions followed novel data analysis  with recalculation of confidence intervals.

Author Response

We would like to thank the reviewers for their insightful comments on our manuscript.  In this revised manuscript, we have made every effort to address the concerns raised by the reviewers. Below are our point-by-point responses to the reviewers’ comments (shown in blue fonts).

Reviewer #3:

The manuscript, devoted to the computer calculation of ligands for intracellular targets, represents a modern direction in the use of high-performance computing to search for ligands for receptors and enzymes. Readers of biological journals are more interested in practical results and reviews that show the influence of a molecule on another molecule, a system, or an organism as a whole. The authors tried to present a comprehensive study, but the good computational part was not fully supported by biological experiments.

Part 3.4. De novo screening... is completely unsupported by experiments and should be removed. If the authors consider the results of this part interesting, then this is a reason to publish another article. This article takes these unsubstantiated predictions out of context.

Response: Thanks for these helpful suggestions. We now removed Part 3.4, and only simply indicated that ChemGPT-cancer may help in future for finding de novo inhibitors of TIPE3 in Discussion (Page 21, Lines 582-603). As suggested, we may publish another paper in the future with additional data support.

  1. Problems exist in the presentation of the material, or rather in the repeated repetition of the same facts in different sections without developing any ideas and comparing with previously published data. I propose to completely remove the first two paragraphs of the Discussion and whole Conclusion. The purpose of the article is not to write down facts several times for memorization, but to briefly convey important results.

Response: Thanks for the helpful suggestion! We have now completely removed the first two paragraphs of the Discussion (Page 21, Lines 517-534) and revised Conclusion in the revised manuscript (Pages 21, Lines 605-617). The manuscript focused only on important findings from this work.

  1. On page 12 Figure S1. K784-8160 exhibits... this should be a second supplementary figure not S1, but I would to recommend to insert it as a part of Figure 4.

Response: We thank the reviewer for raising this important issue and apologize for our oversight. We have moved these figure panels to Figure 4f-4h (Page 13).

  1. Also, Figure S1 on page 9 as important for understand should be included in the main text as well.

Response: We appreciate the reviewer’s suggestion. The focus of our experimental section is to validate the anti-tumor activity of the six potential inhibitors of TIPE3 identified in this work. The relative expression levels of TIPE3 shown in Supplemental Figure 1 (Originally named as Figure S1) in some cell lines, such as HT-29 and Jurkat, have been mentioned in previously published work. Considering that Supplemental Figure 1 is only intended to facilitate the selection of appropriate cell lines for experimental validation, we believe that presenting Supplemental Figure 1 as supplementary material is appropriate.

  1. In Figure 3, the graphs show the data as viability to a concentration in micromoles. This is true for molecules studied, but the graphs show DMSO controls that do not have concentration dependence. My question is why the values for DMSO are flexible, perhaps it was a statistical gap. Are data normalized? Do authors used mathematical statistic methods for comparison.

Response: We thank the reviewer for raising these important questions. We sincerely apologize for any confusion stemming from our presentation and wish to provide the following clarification.

Firstly, our data are not normalized. Secondly, to exclude the effect of DMSO on cell viability, we set up a DMSO control group for each indicated drug concentration (by adding equal volume of DMSO as in the experimental group). Additionally, it is important to emphasize that in Figures 3e and 3f, when the treatment concentration of the small molecule was 100 μM, the corresponding DMSO concentration exceeded 1‰, thus the DMSO control group also showed toxicity to tumor cells. However, please be assured that we applied a consistent method to calculate cell viability across all DMSO-treated and compound-treated groups, and comparisons were conducted using unpaired Student’s t-tests. We have added corresponding supplementary explanations to Page 11, Lines 370-372.

  1. Looks like the data collection wasn't very good. For many tests, n=3, whereas the optimal recommended replicates are 5. In Methods:

2.6. MTT analysis

2.7. Cell viability assay

2.8. Colony formation assay

The repetition of the experiment is indicated as double.

This is not indicated for other tests, but it seems to me that they were also done with low repeatability.

It is necessary to redo experiments with 3-5 repetitions followed novel data analysis with recalculation of confidence intervals.

Response: We sincerely apologize for the insufficient explanation of our experimental design that caused these concerns. Throughout this study, we ensured the vigor and reproducibility of our findings by conducting at least three independent experiments for all experiments, with each experiment comprising at least three biological replicates per dosing group. To ensure clarity and accuracy, we revised our statements on Page 5, Lines 235 and 261 within the Materials and Methods section, and stated the number of times the experiments were performed on Lines 374, 404, 442, and 455.

Round 2

Reviewer 2 Report

Comments and Suggestions for Authors

The manuscript by Chen et al. describes how new inhibitors of the TIPE3 protein were identified using theoretical methods derived from artificial intelligence and molecular mechanics. Some of the small molecules discovered by the authors were tested in vitro. 

The authors have changed the manuscript as suggested. It's my opinion that a reference could be added to the method cited on line 511:

"our newly developed DeepBindGCN_RG/BC algorithms"